# The Combined Assessment of CTC and *ESR1* Status in Liquid Biopsy Samples Enhances the Clinical Value of Prediction in Metastatic Breast Cancer

**DOI:** 10.3390/ijms26052038

**Published:** 2025-02-26

**Authors:** Malgorzata Szostakowska-Rodzos, Ewa A. Grzybowska, Izabella Mysliwy, Renata Zub, Agnieszka Jagiello-Gruszfeld, Maryna Rubach, Aleksandra Konieczna, Anna Fabisiewicz

**Affiliations:** 1Department of Molecular and Translational Oncology, Maria Sklodowska-Curie National Research Institute of Oncology, Roentgena 5, 02-781 Warsaw, Poland; malgorzata.szostakowska-rodzos@nio.gov.pl (M.S.-R.); izabella.mysliwy@nio.gov.pl (I.M.); 2Cancer Molecular and Genetic Diagnostics Department, Maria Sklodowska-Curie National Research Institute of Oncology, Roentgena 5, 02-781 Warsaw, Poland; renata.zub@nio.gov.pl; 3Oncological Clinic, Oncological Mazovian Hospital, al. Solidarności 10, 03-411 Warsaw, Poland; agnieszka.jagiellogruszfeld@gmail.com; 4Cancer Chemotherapy Day Unit, Maria Sklodowska-Curie National Research Institute of Oncology, Roentgena 5, 02-781 Warsaw, Poland; maryna.rubach@nio.gov.pl; 5Department of Breast Cancer and Reconstructive Surgery, Maria Sklodowska-Curie National Research Institute of Oncology, Roentgena 5, 02-781 Warsaw, Poland; aleksandra.konieczna@nio.gov.pl

**Keywords:** breast cancer, liquid biopsy, circulating tumor cells, circulating tumor DNA, *ESR1* mutation, PI3K mutation, droplet digital PCR

## Abstract

Monitoring of metastatic breast cancer (mBC) is an important issue in the clinical management of patients. Liquid biopsy has become a non-invasive method for detecting and monitoring cancer in body fluids. The presence of circulating tumor cells (CTCs) and circulating tumor DNA (ctDNA) in peripheral blood indicates poor prognosis and may contribute to early detection of progression, but assessment of these levels is still not routine clinical management. The main objective of this study was to estimate the frequency and clinical value of the *ESR1* and *PIK3CA* mutations identified in circulating free DNA (cfDNA.) The second goal was to evaluate whether simultaneous evaluation of CTCs and mutation status in cfDNA increases the prognostic value of liquid biopsy. The results of the analysis of the CTC number and *ESR1* and *PIK3CA* mutations in blood collected from 179 patients with metastatic breast cancer show that *ESR1* mutations are more frequent in patients with advanced luminal breast cancer regardless of the type of the treatment. *ESR1* mutations appear primarily during progression, as no mutations were found in primary tumor samples. The main conclusion of the study is that combined assessment of CTCs and *ESR1* status in liquid biopsy may improve the prognostic value of liquid biopsy.

## 1. Introduction

Metastatic breast cancer (MBC) is a treatable but still incurable disease. The majority of diagnosed breast cancers are the luminal type expressing estrogen receptors (ERs) and progesterone receptors (PRs) [1]. About 30–50% of patients will eventually relapse due to resistance to the given treatment [2]. This resistance is a consequence of modifications of ERα at the genetic, regulatory, or protein level that allow tumor growth independent of the presence of estrogen. Resistance often develops as a result of acquiring new mutations in the *ESR1* gene [3,4]. These mutations are rare (up to 3%) in primary tumors but much more abundant in metastatic lesions, where the rate ranges from 5 to 60% [5]. The most common alterations in *ESR1* are point mutations occurring in the ligand binding domain (LBD), in codons 536, 537, 538, and 380 [6,7]. These *ESR1*-LBD mutations result in constitutively activated ERs, causing decreased sensitivity to endocrine treatments [6,8]. For this reason, tracking mutational changes in ERα during treatment has clinical value and may influence therapeutic decisions during treatment.

Another gene frequently mutated in breast cancer is *PIK3CA* (coding the catalytic subunit p110α of phosphatidylinositol 3-kinase [PI3K]). The phosphatidylinositol 3-kinase/protein kinase B/mammalian target of the rapamycin (PI3K/AKT/mTOR) pathway is a pivotal intracellular signaling system, and its hyperactivation is a well-known cause of hormonal treatment failure [9]. Mutations in *PIK3CA* occur mainly in two hotspots at 1047aa and 545aa, accounting for around 70% of all mutations [10,11]. They lead to constitutive activation of PI3K, which has been proposed as a mechanism for endocrine resistance [10].

Due to the heterogeneity of cancer and its dynamic development, recognition of molecular mechanisms responsible for cancer evolution is a challenge. Recently, liquid biopsy has emerged as a noninvasive method for detecting and monitoring cancer in body fluids rather than tumor tissue. Cancer cells release CTCs, ctDNA, RNA (mRNA and micro-RNA), and extracellular vesicles (EV), which can be detected in peripheral blood collected repeatedly during a patient’s treatment [12,13,14]. Unlike tissue biopsy, liquid biopsy can be taken simultaneously with routine blood tests at any stage of the disease. Therefore, a new field of oncology has emerged, focusing on the components of analysis of metastatic tumors circulating in the blood, mainly CTCs and ctDNA. Research in this area will help to better track cancer progression and tailor treatment.

The presence of cfDNA in the blood has proven prognostic significance. Dawson et al. demonstrated that ctDNA is a specific and highly sensitive biomarker in MBC [15], outperforming CTCs in detection frequency and correlation with tumor burden. This was confirmed by other studies [16,17]. Genomic analysis of ctDNA has begun to be incorporated into the clinical management of patients with advanced cancer. The mutational analysis of *ESR1* and *PIK3CA* in cfDNA from MBC patients has been recognized as an important tool for the assessment of the response to treatment and drug efficacy [18,19,20] and has been tested in clinical trials. Mutations in *ESR1* and *PIK3CA* did not show an effect on PFS (progression-free survival) and OS (overall survival) in the MONARCH study [21] but were associated with worse survival in other studies, e.g., the BOLERO-2 trial and the SAFIR02 trial [22,23]. Mutations in other functionally important genes in MBC were tested in the PALOMA-3 trial and PEARL trial, in which *TP53* mutations and *FGFR1* amplifications were associated with worse outcome regardless of treatment [24,25].

CTCs have been identified as an independent prognostic factor for PFS and OS in the adjuvant, neoadjuvant, and metastatic settings [26,27]. CTC research has also been used as a tool in studying breast cancer heterogeneity [28]. CTC detection and its longitudinal analysis [29,30,31] are still not a clinical standard, and its value as a predictor of disease progression has not yet been established, although many studies suggest that it could be a promising prognostic tool for clinicians [32,33,34]. Until now, several studies have attempted to combine ctDNA information with CTCs [35,36,37], although they mainly focus on cfDNA levels or compare the mutational status of cfDNA and gDNA in CTCs.

The general objective of this study was to evaluate the frequency and clinical value of the *ESR1* and *PIK3CA* mutations identified in cfDNA and to compare these results with primary tumor samples to evaluate if the mutation was originally present in the tumor or if it occurred during metastasis. The other goal was to assess whether the simultaneous evaluation of the status of CTCs and mutational status in cfDNA might strengthen the prognostic value of liquid biopsy. Our work demonstrates that combining the two liquid biopsy approaches results in a better prognosis.

## 2. Results

### 2.1. Patients’ Characteristics

In total, 179 patients were enrolled in the study. The clinical characteristics of the patients are summarized in Table 1 and Figure 1. The median follow-up was 53.1 months. The median age of the patients was 63 years at the beginning of the study. Approximately 70% of the patients were characterized with bone metastases. Most of the histological subtypes identified in the primary tumor sample were NST. Most of the patients were treated with radiotherapy and HTH + CHTH + CDK4/6.

### 2.2. Mutations in cfDNA

In general, mutations in *ESR1* or *PIK3CA* were found in ~63% of the patients. *ESR1* mutations were more frequent, as they were found in 53.63% of patients, while *PIK3CA* mutations in *PIK3CA* were found in 26.82% of patients (Table 2, Figure 2). Around 15% of the patients were characterized with double mutations in the *PIK3CA* gene and the *ESR1* genes. One patient had double *PIK3CA* mutations, occurring in p.E545K and p.H1047R hotspots.

For 59 patients, an additional analysis of the mutation status was performed in FFPE (formalin fixed, paraffin embedded) samples as representative samples of a primary tumor. This enabled us to compare the mutational status in primary tumors vs. liquid biopsy. No patient was found to have a mutation in *ESR1* in the primary tumor sample, while ~12% were identified with *PIK3CA* mutations. From this group, ~54% of patients with *ESR1* mutations and ~20% with *PIK3CA* mutations were identified on liquid biopsy (Figure 2). Interestingly, we showed that all *ESR1* mutations were identified only in cfDNA samples, while for *PIK3CA* mutations, the mutation gain during progression was observed only in a few patients. These results highlight that *ESR1* mutations develop under the selective pressure of endocrine treatments and might be associated with cancer progression. Therefore, for further validation of the clinical value of cfDNA mutational status, we analyzed data considering only *ESR1* mutational status.

### 2.3. Clinical Value of Liquid Biopsy

CTC evaluation

For 96 patients out of a total of 179 patients subjected to ctDNA analysis, an additional CTC evaluation was performed. The characteristics of this group are shown in Table 3. The median age in this group was ~65. CTCs were detected in 36% of the patients, and ≥5 CTCs were found in ~16% of the patients. Patients with ≥5 CTCs detected were characterized with a significantly lower median survival (Figure 2A and Figure 3B). For all patients, an additional evaluation of *ESR1* mutational status in cfDNA was performed. To estimate the clinical value of the liquid biopsy, additional analysis was performed for combined liquid biopsy markers as predictors of OS. For patients with ≥5 CTCs and *ESR1* mutation in cfDNA material, the median survival was significantly lower than for other patients (11.1 months, compared to 44.3 for patients ≥5 CTCs and N/A for patients with <5 CTCs) (Figure 3C,D). These results highlight that the simultaneous evaluation of liquid biopsy markers might improve the prognostic value of liquid biopsy during treatment.

Multivariable COX proportion and hazard regression analysis

To further confirm the clinical value of the simultaneous assessment of the CTCs and the *ESR1* mutation, we performed the multivariable Cox proportional hazard regression. The clinicopathological data used for the Cox multivariable model are listed in Table 4 with their reference levels.

Interestingly, the ≥5 CTCs and *ESR1* status in cfDNA alone were found to not be significant factors in the univariable and multivariable Cox analysis for the overall survival prognosis. However, when combined, these markers were found to greatly improve the prognostic value of liquid biopsy. The simultaneous presence of ≥5 CTCs and *ESR1* mutation in liquid biopsy was found to be a strong predictive factor of OS in univariable (HR = 3.496; 95% CI 1.173–8.484; *p*-value < 0.05) and multivariable (HR = 3.538; 95% CI 1.126–9.403; *p*-value < 0.05) analyses (Table 5, Figure 4). Furthermore, these results highlight that combining standard liquid biopsy approaches strongly improves the clinical effectiveness of liquid biopsy as a predictor of OS. Additionally, correlation analysis between *ESR1* status and CTC status have shown the absence of correlation (Appendix A), supporting the assumption of the independence of these variables.

In a multivariate Cox hazard regression model, we also observed higher HRs for the lobular subtype and combined treatment (HTH + CHTH; HTH + CDK4/6; HTH + CHTH + CDK4/6) (Figure 4). These results might be associated with a low number of patients with the lobular subtype (n < 10), high advancement, and disease aggressiveness for the patients treated with combined therapy.

## 3. Discussion

Liquid biopsy is one of the most dynamically improving fields in current clinical science. The evaluation of liquid biopsy potential as a prognostic tool is usually done using cfDNA mutational analysis or CTC number counts.

The evaluation of the genetic status of *ESR1* and *PIK3CA* in cfDNA of patients with MBC is recognized as having prognostic and predictive value [38,39,40,41]. Plasma *PIK3CA* ctDNA specific mutation detected by next generation sequencing is associated with clinical outcomes in advanced breast cancer.

The results of the research carried out in this work confirm previous findings that *ESR1* mutations are more frequent in metastatic luminal breast cancer patients irrespective of the type of therapy (endocrine or chemotherapy-based treatments) [38,42] and reveal that pathogenic *ESR1* mutations appear mainly during progression, most likely as a result of the selective pressure of endocrine treatments, as no mutations were found in primary tumor samples. *PIK3CA* mutations are more frequent in primary tumors, and most of them remain present in the advanced stages, but only a few new mutations appear during the metastatic process. These results and the statistical analysis of mutational changes in both genes and their impact on survival (Figure 3) suggest that *ESR1* mutations in cfDNA are better suited to be a prognostic tool in MBC. However, our results on the impact of cfDNA mutations were on the borderline of significance (*ESR1*) or insignificant (*PIK3CA*) in Kaplan–Meier survival analysis. The clinical relevance of the *ESR1* mutations detected in cfDNA evaluated by Kaplan–Meier analysis has been reported several times, with different significance levels [43,44,45,46]. The borderline significance of our analysis may result from the relative heterogeneity of our group of patients in terms of the observation period, so in some patients, resistance-conferring mutations may not be present because there was not enough time for the evolution of the resistance.

However, the results point to the conclusion that this prognostic value of liquid biopsy should be further improved, possibly by combining it with CTC enumeration. This could be achieved by evaluating both values from the same blood sample, which makes it relatively easy to implement in clinical practice.

CTCs are significant prognostic markers in metastatic breast cancer patients. The prognostic value of CTCs has been widely studied, and the presence of ≥5 CTCs was found to be a negative prognostic predictor of OS and PFS [47,48,49]. Longitudinal studies of serially collected samples were reported to improve the prognostic power of CTC enumeration in metastatic breast cancer but require a more organized collection schedule [31,50,51].

In the current study, the presence of CTCs with the ≥5 cutoff was confirmed to be significant. Subsequently, we tested whether the combination of CTC numbers and *ESR1* mutational status in blood samples from advanced breast cancer patients can improve the clinical value of a single liquid biopsy. The question of the benefits of combining these two markers has been addressed in several reports, but the authors evaluated the level of cfDNA, its integrity, or a whole profile of genomic alterations and did not evaluate the presence of specific mutation(s) in one gene [35,36,37,52,53].

There are several limitations to our study. First, our findings may have been influenced by heterogeneity of the group, which was not uniformly advanced in the disease. Second, we do not compare mutational status of *ESR1* between cfDNA and CTCs, and some reports suggest that the frequency of mutation might be different [54]—however, it could be argued that the cited study had a small sample size and did not achieved significance.

Overall, our findings suggest that the presence of plasma *ESR1* mutations in addition to the ≥5 CTC number is unfavorable in the long-term prognosis for these patients and adds additional evidence that early detection of mutation may be clinically helpful for the prediction of treatment efficacy. Additionally, this will help select a specific group of patients who will benefit from a change in treatment. A combined approach would also represent optimal, efficient use of the liquid biopsy sample obtained in one collection.

## 4. Materials and Methods

### 4.1. Patients Samples

Blood samples were collected from 179 patients with advanced luminal breast cancer progressing under hormone therapy. Blood collection (9 mL) was carried out once during the treatment follow-up in the Maria Sklodowska-Curie National Institute of Oncology. The selection of patients was carried out by experienced clinicians from the Department of Breast Cancer and Reconstructive Surgery of the Maria Sklodowska-Curie National Research Institute of Oncology. Patients were included for this study between June 2018 and December 2022. The inclusion criteria for the patients were breast cancer with ongoing hormonal treatment, age ≥ 18, and identification of distant metastases. All participants signed an informed consent. Overall, 179 patients were enrolled: 179 patients with cfDNA mutation evaluation and 96 patients with CTCs and cfDNA evaluation.

### 4.2. Isolation and Preparation of cfDNA

Plasma samples from patients were isolated using the QIAamp Circulating Nucleic Acid Kit (Qiagen; Hilden, Germany) according to the manufacturer’s protocol. The amount of isolated cfDNA was measured using the Quantus system (Promega, Walldorf, Germany) using the QuantiFluor ONE dsDNA system (Promega; Walldorf, Germany). The isolated and measured cfDNA was further used for ddPCR analysis.

### 4.3. ddPCR Analysis

To test the abundance, ddPCR analysis was performed using the BioRad QX200 Droplet ddPCR system according to the protocol of Schiavon et al. [18]. Probes for *ESR1* mutations L536R, Y537S/C, and D538G (Table 6) were purchased from Merck (Rahway, NJ, USA). Reactions were run in multiplexes: L536R with Y537C and Y537S with D538G. For *PIK3CA* E545K and H1047R mutations, BioRad specific ddPCR Mutations Assays were used (Hercules, CA, USA). The plates were read on a BioRad QX200 droplet reader with BioRad QuantaSoft v1.6.6.0320 software (Hercules, CA, USA).

### 4.4. Post-Analysis for ddPCR

The estimation of the false positive rate was determined by performing 5 experiments for each assay using WT-only samples, where total amounts of detected mut-positive droplets determined thresholds above which positive droplets in patient samples were to be considered true positive. For each patient, plasma was analyzed in duplicate. Therefore, the PCR results of the patient samples were based on the mean estimated target DNA concentrations (copies/μL) in the merged wells, automatically calculated by the manufacturer’s software. Correction for false positivity was made by subtracting the number of mut-positive droplets detected in the false positive assessment experiments. The mutant allele frequency (MAF) was defined as the number of mut-positive droplets in the total droplet amount (mut-positive and wt-positive). Samples were considered positive if mutation was confirmed in the FFPE sample and mut-positive droplets were found.

### 4.5. CTC Assessment

After the collection, the blood samples were centrifuged at 2500× *g* for 15 min. The plasma was collected for the cfDNA analysis, while the buffy coat was transferred to a new 15 mL tube. The CTC assessment with the CytoTrack system was done according to the previously established protocol [34]. The collected cells were stained with the following: Alexa Fluor 488-conjugated pancytokeratin (pan-CK) antibody (1:25) (ThermoFisher Scientific, Waltham, MA, USA), APC conjugated CD45 antibody (1:10) (ThermoFisher Scientific, Waltham, MA, USA), PE conjugated EpCAM antibody (1:10) (ThermoFisher Scientific, Waltham, MA, USA), and 4,6-diamidino-2-phenylindole (DAPI) (Sigma-Aldrich, St Louis, MO, USA) (1:1000). The focus plan for scanning was obtained based on the DAPI stain, at eight places on the disc. Scanning was performed with a 488 nm argon-neon laser, in a spiral pattern with a bandwidth of 10 μm, for 5 min. All signals from the Alexa Fluor 488 emission channel (pan-CK) were recorded and listed in the hotspot table (positive events). The criteria for CTC identification were established as follows: nearly round size with ≥6 µm diameter, visible nucleus, pan-CK signal, and CD45 negative. The clusters were defined as follows: group of ≥3 cells, with at least 3 visible nuclei in the DAPI channel, and with at least 3 cells identified as CTCs.

### 4.6. FFPE Analysis

FFPE samples obtained from the Department of Cancer Pathomorphology in the Maria Sklodowska-Curie National Research Institute of Oncology were cut 10 µm thin, and up to 8 sections were used for DNA isolation. DNA was isolated using the QIAamp DNA FFPE Tissue Kit (Qiagen, Hilden, Germany) according to the manufacturer’s instructions. The purity of the isolated genetic material was verified on a NanoDrop spectrophotometer (ThermoFisher, Waltham, MA, USA). Only pure DNA with a concentration of at least 50 ng/µL was used for sequencing. Samples were amplified using GoldTaq Polymerase (Applied Biosystems, Waltham, MA, USA) and GeneAmp PCR System 9700 Thermal Cycler (Applied Biosystems, Waltham, MA, USA). PCR products were sequenced using BigDye™ Terminator v3.1 Cycle Sequencing Kit (ThermoFisher, Waltham, MA, USA) and ABI Prism 3130xl Genetic Analyzer (ThermoFisher Waltham, MA, USA). The primers’ sequences are shown in the Table 7.

### 4.7. Statistical Analysis

Categorized quantitative data at different time points were compared using the Mann–Whitney U test or, if there were more than two categories, the Kruskal–Wallis test. The primary end point was overall survival (OS). OS was defined as the time from blood collection to death from any cause. If an outcome was not reached during the observation time, the variables were censored. Kaplan–Meier plots and log-rank tests were used to illustrate and compare survival between subgroups. Univariable and multivariable hazard ratios (HRs) for selected potential predictors of OS were determined by Cox proportional hazards regression. The fit was measured using the Harrell C index, and the fit of the nested prognostic models was compared using the logarithmic likelihood ratio test (G squared). All data were analyzed using GraphPad Prism 9.

## Figures and Tables

**Figure 1 ijms-26-02038-f001:**
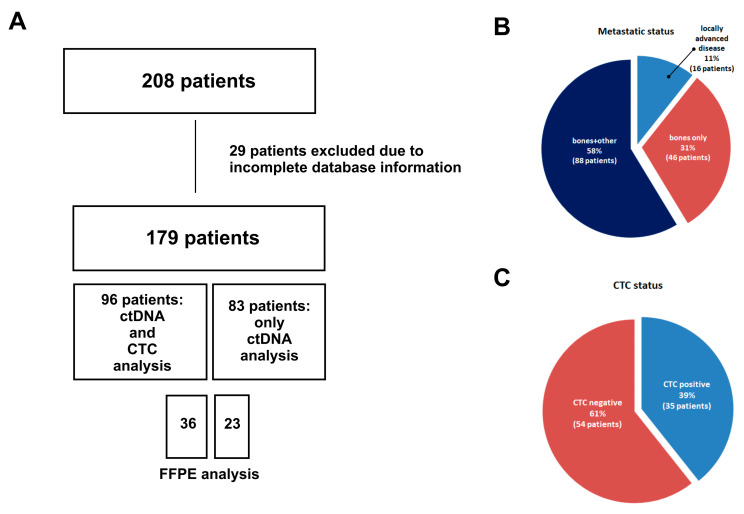
Flowchart with the overall patient data: (**A**) flowchart of the study design; (**B**) metastatic status of patients included in the studies, divided into bone metastasis only, bones and other lesions, and locally advanced; (**C**) CTC status of patients included in the studies.

**Figure 2 ijms-26-02038-f002:**
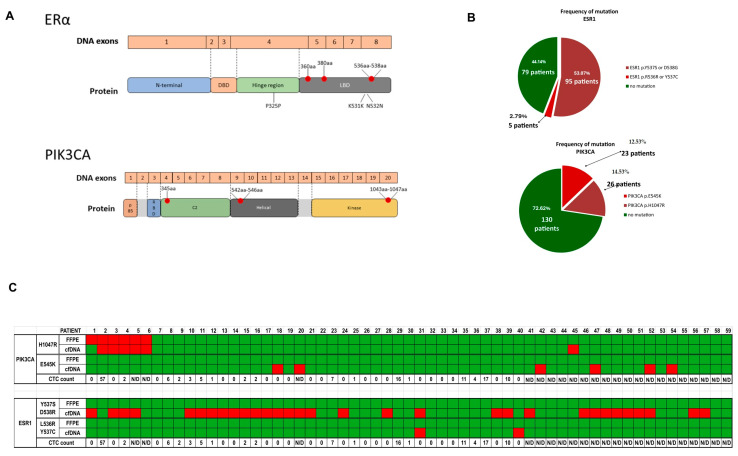
The mutational status in cfDNA of patients included in the study: (**A**) The graphical representation of mutational hotspots and main mutations detected in this study in *PIK3CA* and *ESR1* genes. (**B**) The frequency of mutations in *PIK3CA* and *ESR1* in cfDNA; data from 179 patients’ samples. (**C**) The incidence of mutations in *PIK3CA* and *ESR1* genes in samples from 59 patients for whom the analysis of FFPE and cfDNA mutational status was done; for patients with CTCs detected, the number of CTCs was also included; N/D is for patients without CTCs. red = mutation; green = no mutation.

**Figure 3 ijms-26-02038-f003:**
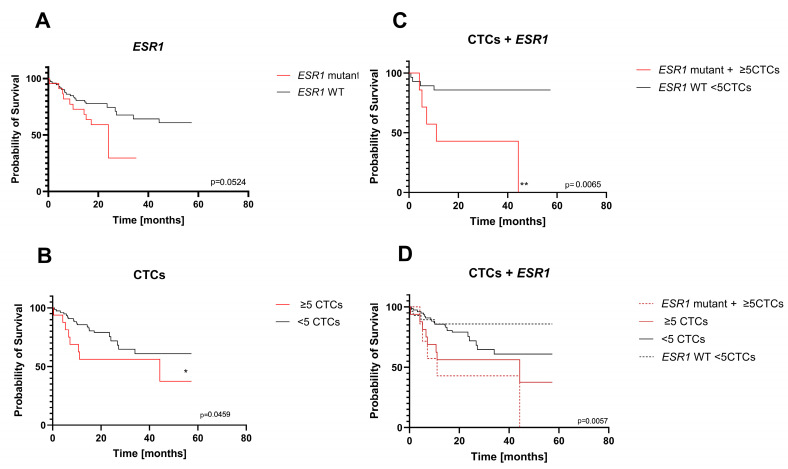
The Kaplan–Meier survival analysis for overall survival (OS) for a group of 96 patients with CTCs and *ESR1* mutation. (**A**) Patients divided according to *ESR1* mutational status; mutation detected—red line, wild-type only detected—black line. (**B**) Patients divided according to the CTC status: ≥5 CTCs—red line, <5 CTCs—black line. (**C**) Patients divided according to combined CTCs and *ESR1* status: ≥5 CTCs and *ESR1* mutation—red line, <5 CTCs and wild-type *ESR1*—black line. (**D**) Patients divided according to combined CTCs and *ESR1* status: ≥5 CTCs—red line, <5 CTCs—black line, ≥5 CTCs and *ESR1* mutation—dashed red line, <5 CTCs and wild-type *ESR1*—dashed black line. The *p*-values for survival analysis in log-rank test were stated at the bottom right corner of each graph: *—*p*-value < 0.5; **—*p*-value < 0.01.

**Figure 4 ijms-26-02038-f004:**
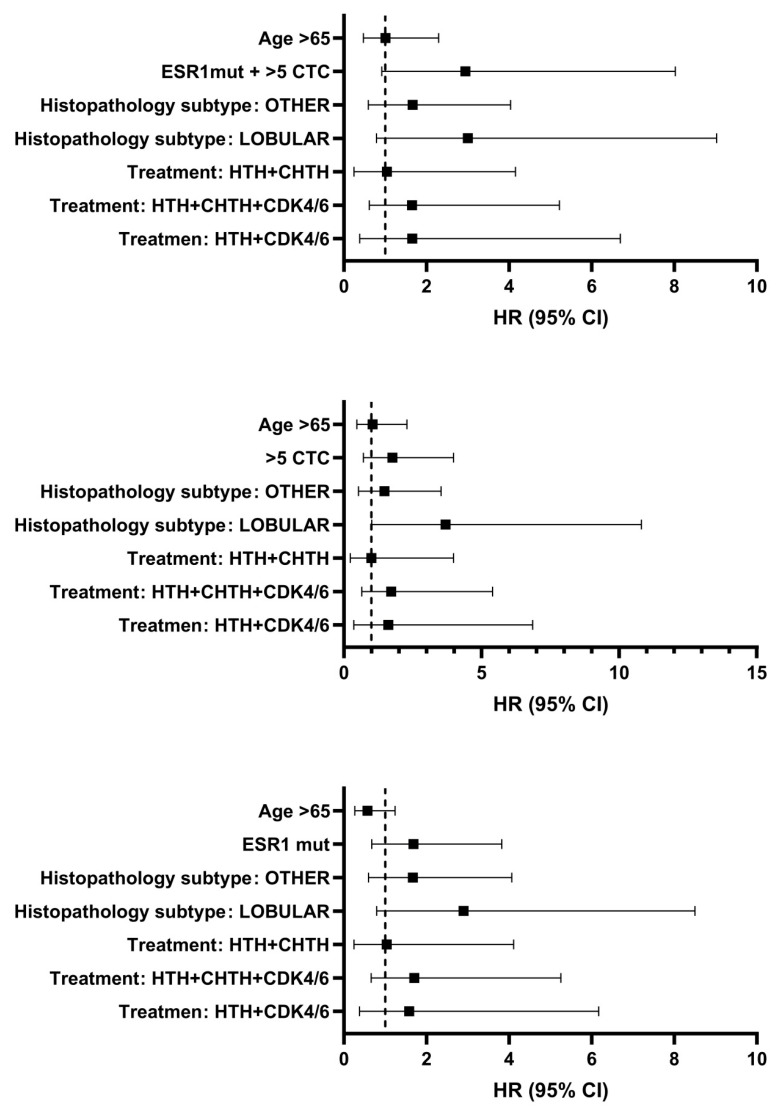
The graphical representation of the hazard ratios (HRs) and confidence intervals (95% CI) for multivariable Cox proportional hazard regression model. The dashed black line represents HR = 1. HR for the combined CTCs ≥ 5 count and the presence of *ESR1* mutation (HR = 3.538; 95%CI 1.126–9.403; *p*-value 0.0172) is significantly higher than HRs for CTC count and *ESR1* mutation alone (HR = 1.814; 95%CI 0.7310–4.124; *p*-value ≥ 0.05, and HR = 0.5616; 95%CI 0.2686–1.165; *p*-value ≥ 0.05, respectively).

**Table 1 ijms-26-02038-t001:** Clinical characteristics of patients.

Variables	Number of Patients
**Age**	
<63	89
≥63	90
**HER2 status**	
HER2+	5
HER−	161
N/D	13
**No of meta sites**	
1	83
2	49
≥3	47
**Meta sites**	
Bones	133
Liver	67
Lung	58
Other	81
**Hitological subtype**	
NST	133
Lobular	16
Other	30
**Treatment**	
HTH	32
HTH + CHTH	36
HTH + CDK4/6	26
HTH + CHTH + CDK4/6	85
**Radiotherapy**	
RTH+	100
RTH−	79

**Table 2 ijms-26-02038-t002:** Frequency of the mutations identified in the cfDNA material.

Mutation	Frequency	No of Patients with Mutated Samples
***PIK3CA*** p.E545K	12.85%	23
***PIK3CA*** p.H1047R	14.53%	26
***ESR1*** p.Y537S or D538G	53.07%	95
***ESR1*** p.R536R or Y537C	2.79%	5

**Table 3 ijms-26-02038-t003:** Characteristics of patients evaluated for CTCs.

Variables	Number of Patients
**Age**	
<65	50
≥65	46
**HER2 status**	
HER2+	5
HER−	86
N/D	5
**No of meta sites**	
1	58
2	27
≥3	21
**Meta sites**	
Bones	70
Liver	32
Lung	26
Other	36
**Hitological subtype**	
NST	75
Other	21
**Treatment**	
HTH	22
HTH + CHTH	15
HTH + CDK4/6	14
HTH + CHTH + CDK4/6	45
**Radiotherapy**	
RTH+	46
RTH−	50
***ESR1* status**	
***ESR1*** mutation	58
***ESR1*** WT	38

**Table 4 ijms-26-02038-t004:** The clinical variables used for the Cox multivariable model.

Clinical Variable	Reference Level
Treatment	HTH
Histopathological subtype	NST
Age	<65

**Table 5 ijms-26-02038-t005:** The results of Cox proportional hazard regression analyses.

**Univariable Analysis**			
**Variable**	**HR**	**95 CI**	***p*-Value**
***ESR1*** mutation	0.5832	0.2853–1.186	0.1339
≥5 CTCs	1.775	0.7423–3.821	0.1636
***ESR1* mut + ≥5 CTCs**	**3.496**	**1.173–8.484**	**0.0113**
**Multivariable analysis**			
**Variable**	**HR**	**95 CI**	***p*-value**
***ESR1*** mutation	0.5616	0.2686–1.165	0.1197
≥5 CTCs	1.814	0.7310–4.124	0.1714
***ESR1* mut + ≥5 CTCs**	**3.538**	**1.126–9.403**	**0.0172**

**Table 6 ijms-26-02038-t006:** Sequence of probes and starters used for the assessment of the *ESR1* mutation status.

Mutation	Probe Sequence	Primer Sequence	
L536R	[6FAM]TGGTGCCCCGCTATGACC[BHQ1]	F	F 5′AGGCATGGAGCATCTGTACA3′
		R	5′TTGGTCCGTCTCCTCCA3′
Y537S	[6FAM]TGGTGCCCCTCTCTGACCT[BHQ1]	F	F 5′AGGCATGGAGCATCTGTACA3′
		R	5′TTGGTCCGTCTCCTCCA3′
D538G	[6FAM]CCCTCTATGGCCTGCTGCT[BHQ1]	F	F 5′AGGCATGGAGCATCTGTACA3′
		R	5′TTGGTCCGTCTCCTCCA3′
Y537C	[6FAM]TGCCCCTCTGTGACCTGCT[BHQ1]	F	F 5′AGGCATGGAGCATCTGTACA3′
		R	5′TTGGTCCGTCTCCTCCA3′
WT ***ESR1***	[HEX]TGGTGCCCCTCTATGACCTG[BHQ1]	F	F 5′AGGCATGGAGCATCTGTACA3′
		R	5′TTGGTCCGTCTCCTCCA3′

**Table 7 ijms-26-02038-t007:** Sequences of the primers used for Sanger sequencing.

Gene	Forward Primer	Reverse Primer	Product Length
***ESR1* exon 8**	5′-TCTGTGTCTTCCCACCTACAGT-3′	5′-ATGCGATGAAGTAGAGCCCG-3′	200 bp
***PIK3CA* exon 9**	5′-AGCTAGAGACAATGAATTAAGGGA-3′	5′-TCCATTTTAGCACTTACCTGTGAC-3′	130 bp
***PIK3CA* exon 20**	5′-AACTGAGCAAGAGGCTTTGGA-3′	5′-CAATCGGTCTTTGCCTGCTG-3′	200 bp

## Data Availability

The original contributions presented in this study are included in the article/Appendix A. Further inquiries can be directed to the corresponding authors.

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
