# Peer review of "The Combined Assessment of CTC and ESR1 Status in Liquid Biopsy Samples Enhances the Clinical Value of Prediction in Metastatic Breast Cancer"

_ijms, 2025, doi:10.3390/ijms26052038_

Round 1
Reviewer 1 Report
Comments and Suggestions for Authors
The manuscript by Szostakowska-Rodzos is a highly interesting research article that explores the role of ESR1 gene mutation in CTCs and cfDNA as a liquid biopsy. my comments:
- The phrase "liquid biopsy" appears twice in the title—consider revising to avoid repetition.
- A more precise and consistent use of abbreviations and punctuation is required. A thorough re-reading of the entire manuscript is necessary to ensure proper punctuation usage and that all abbreviations are defined upon their first mention.
- Gene names should always be written in italics. Additionally, italics should not be used inappropriately in other parts of the text (e.g., in the Introduction).
- The Introduction is redundant and should be restructured for clarity. The role of CTCs in breast cancer is well established—consider citing this article: [10.3390/biomedicines9091242].
- In the Introduction, the authors mention alpelisib—are there patients in this study treated with this drug? If not, clarify its relevance.
- Ensure consistency in terminology: use either advanced breast cancer or metastatic breast cancer, but not both interchangeably.
- Section 2.2: Report exact percentages along with the corresponding number of patients (e.g., the number of patients with a specific mutation). The same applies to tables—providing both absolute numbers and percentages improves interpretability. This is particularly important for Table 2. Avoid using the tilde symbol (~).
- Section 2.3: Clarify the cut-off value for CTCs—should it be ">5 CTCs" or "≥5 CTCs"? Ensure consistency throughout the manuscript, including in Figure 3.
- Materials and Methods: The authors should specify more clearly how and when blood samples were collected.
- In table captions, all abbreviations used should be explicitly defined.
- A recently published article [10.1002/1878-0261.13787] is relevant to this study—I suggest citing it and comparing the results accordingly.
Reviewer 2 Report
Comments and Suggestions for Authors
This study explored the mutation of ESR1 and PIK3CA in circulation cfDNA and investigated the prognostic potential of combining CTCs with the mutation status of ESR1. It's an interesting study. It would be helpful to readers if the following points could be addressed.
- The limitations of this study should be addressed.
- Table 2 and Figure 2 require further clarification, particularly regarding the number of mutated samples. I assumed that the red square in Figure 2A denotes mutation occurrence; however, my frequency calculations did not match yours. Additionally, since all 59 samples were obtained from metastatic breast cancer patients, it's unclear how these can be considered representative of primary tumors.
- The detection of circulating tumor cells (CTCs) needs to be described in more detail, as there are no specifics provided in your reference 34. Please include your criteria for defining the cut-off value for CTCs (>5 CTCs) in the Materials and Methods section and explain the rationale for this choice.
- Have you checked the correlation between ESR1 mutations and CTC numbers among the 96 tested samples? Do these mutations co-occur within the same patients?
- Figure 4 is somewhat confusing. The only apparent difference among the three panels is the second line. It appears that the combination of HTH with other treatments is associated with worse outcomes compared to HTH alone. How can this be explained?
Round 2
Reviewer 1 Report
Comments and Suggestions for Authors
The authors addressed properly all my concerns. The manuscript can be accepted in current form.
Reviewer 2 Report
Comments and Suggestions for Authors
Comments were addressed.